# Reduction of Blood Oxidative Stress Following Colorectal Cancer Resection

**DOI:** 10.3390/cancers16203550

**Published:** 2024-10-21

**Authors:** Katsuji Sawai, Takanori Goi, Youhei Kimura, Kenji Koneri

**Affiliations:** First Department of Surgery, University of Fukui, Fukui 910-1193, Japan; tgoi@u-fukui.ac.jp (T.G.); y-kimura@kimura-hospital.jp (Y.K.); koneri@u-fukui.ac.jp (K.K.)

**Keywords:** colorectal cancer resection, oxidative stress, reactive oxygen metabolite derivatives, biologic antioxidant potential

## Abstract

Colorectal cancer is a major global health burden, with surgical resection being the standard treatment aimed at curative tumor removal. Oxidative stress is crucial in colorectal cancer progression and prognosis. This study hypothesizes that the physical removal of colorectal cancer, a primary source of oxidative stress, would reduce blood levels of reactive oxygen metabolite derivatives (d-ROMs), a marker of oxidative stress. This study included 123 patients who underwent radical resection for colorectal cancer. d-ROM levels were measured before and after surgery. The clinicopathological analysis showed a correlation between preoperative d-ROM levels and tumor size. This study confirmed a significant reduction in d-ROM levels following tumor resection. The d-ROM ratio before and after tumor resection was significantly higher in cases with positive lymph node metastasis and larger tumor size. These results suggest that d-ROM levels could serve as a valuable biomarker for monitoring tumor burden in patients with colorectal cancer.

## 1. Introduction

Colorectal cancer (CRC) is a major global health burden, ranking as the third most commonly diagnosed cancer and second leading cause of cancer-related deaths worldwide. In 2020 alone, approximately 1.9 million new cases were diagnosed and 935,000 deaths were reported globally [1]. The standard treatment for CRC typically involves surgical resection aimed at the curative removal of the tumor along with the adjacent lymph nodes [2].

Oxidative stress (OS) is defined as an imbalance between reactive oxygen species (ROS) production and antioxidant defenses. This imbalance can arise from either excessive ROS production or a deficiency in the antioxidant system [3,4,5]. OS plays a role in the initiation, progression, metastasis, and prognosis of CRC, and is integral to cellular signaling processes. However, excessive OS can lead to DNA damage and the oxidation of critical lipids and proteins. These changes disrupt intracellular signaling, cause cellular mutations, impair cellular functions, and promote carcinogenesis [6,7]. Transformation from colorectal adenoma to carcinoma involves the activation of oncogenes, inactivation of tumor suppressor genes, and DNA methylation. These mutations are primarily induced by OS and have significant implications for cancer development [8,9].

The effect of OS on metastasis and tumor growth includes the induction of lipid peroxidation within the mitochondria, leading to the expression of cell cycle-activating proteins, cytokines, growth factors, and adhesion molecules that promote tumor growth and metastasis [10,11,12,13,14,15,16,17]. Reports on OS and CRC prognosis indicate that a high OS in CRC cases is associated with poor outcomes.

OS promotes telomerase activity in tumor cells and increases resistance to chemotherapy and radiotherapy, which may reduce survival rates [18,19,20,21,22,23,24]. Additionally, CRC tissues exhibit higher OS and antioxidant capacity than the surrounding normal tissues [25,26,27,28,29,30,31].

This is attributed to the strong inflammation observed in tumor tissues, where various cytokines are secreted, stimulating neutrophils and monocytes within the tumor and leading to a rapid increase in OS [18,19]. In addition, because of mitochondrial dysfunction, metabolic changes, and frequent genetic mutations, ROS production is significantly increased in cancer cells, resulting in a marked accumulation of oxidized proteins, DNA, and lipids [32,33]. The increase in antioxidant capacity is reported to result from the activation of genes coding for antioxidant enzymes as a cellular adaptation to OS [27]. Despite the observed close relationship between OS and CRC, few studies have examined changes in OS and antioxidant capacity following the surgical resection of CRC.

Recently, methods such as the measurement of reactive oxygen metabolite derivatives (d-ROMs) and biologic antioxidant potential (BAP) have been reported for the quantitative and stable measurement of OS and antioxidant capacity in blood [34]. We previously reported that blood d-ROM levels are associated with advanced tumor size and that high blood d-ROM levels are associated with a poorer prognosis in CRC cases [22,23,24].

In this study, we tested the hypothesis that physical removal of CRC, which is the source of OS, would reduce d-ROM levels. If a decrease is observed, d-ROM levels may reflect tumor burden and could be a valuable tumor marker.

## 2. Materials and Methods

### 2.1. Study Population

This study included 123 patients who underwent a radical resection for CRC at our institution between 2020 and 2023. Patients with synchronous or metachronous cancers, inflammatory bowel disease, immunosuppressive diseases, or severe medical illness, and those who underwent preoperative chemoradiation therapy, emergency surgery, or hemodialysis, were excluded. We collected data on patient age, sex, tumor location, tumor size, depth of tumor invasion, lymph node metastasis, and stage. The histopathological and clinical staging of tumors were assessed based on the TNM classification.

### 2.2. Measurement of d-ROMs and BAP Levels

We measured the d-ROM and BAP levels before and one month after surgery using blood samples collected during regular examinations.

Serum d-ROM and BAP levels were measured using a Free Radical Elective Evaluator system (FREE Carpe Diem, Wismerll Co., Ltd., Tokyo, Japan) that included a spectrophotometric device. Reader and measurement kits (d-ROMs and BAP test, Wismerll Co., Ltd.) were optimized for the FREE Carpe Diem System according to the manufacturer’s protocol. Briefly, in the d-ROM test, 20 µL of serum sample and 1 mL of buffered solution were carefully mixed in a cuvette, followed by the addition of 20 µL of chromogenic substrate. After thorough mixing, the cuvette was placed in the analyzer’s thermostatic block for 5 min at 37 °C, and absorbance was measured at 505 nm. Results were reported in arbitrary units (U.CARR), with each unit corresponding to 0.8 mg/L of hydrogen peroxide. The reference range was 250–300 U.CARR, and levels ≥ 300 U.CARR indicated serum OS due to excessive free radical production [5,35,36,37].

Briefly, in the BAP test, 10 µL serum samples were added to a 1 mL assay mixture, and the amount of trivalent iron deoxidized over 5 min was measured in µmol/L. When FeCl_3_ is dissolved in a colorless solution with a chelation acid derivative, it turns red due to Fe^3+^ ions. This red color is decolorized by the reduction of Fe^3+^ to Fe^2+^ ions caused by the antioxidant activity of the plasma. The antioxidant potential of plasma is evaluated by measuring the degree of decolorization with a spectrophotometer. The normal BAP value in healthy subjects is >2200 µmol/L [37].

### 2.3. Assessment of d-ROM and BAP Levels

The relationship between preoperative d-ROM levels, BAP levels, and clinicopathological factors was examined, as well as the relationship between preoperative and postoperative d-ROM and BAP levels. Additionally, the changes in d-ROM and BAP values before and after tumor resection in each stage were investigated. We also analyzed the d-ROM and BAP ratios (i.e., the ratio of d-ROM and BAP levels before and after treatment) and their relationship with clinicopathological factors.

### 2.4. Statistical Analysis

Continuous variables were analyzed using a Mann–Whitney U test and are expressed as the median (interquartile range). We conducted a multiple linear regression analysis to identify the factors that might influence preoperative d-ROM levels, preoperative BAP levels, d-ROM ratio, and BAP ratio. The factors examined included age, sex, tumor location, tumor size, serosa invasion, and lymph node metastasis. After establishing the final model, we calculated residuals and created a normal probability plot to evaluate the normality of these residuals. Spearman’s correlation coefficient analysis was employed to explore the relationships between d-ROM and BAP levels. All statistical analyses were conducted using IBM SPSS software version 21.0 (IBM Japan, Ltd., Tokyo, Japan), with significance set at *p* < 0.05.

## 3. Results

### 3.1. Patient Characteristics

The baseline demographic and clinicopathologic data of 123 patients with CRC are summarized in Table 1. In total, 22 cases had tumor depth T1, 20 had T2, 25 had T3, and 56 had T4. Additionally, three cases had stage 0, 38 had stage I, 40 had stage II, and 42 had stage III. Patients underwent curative resection with lymph node dissection at our institution between 2020 and 2023. Patients who had postoperative complications and those who received adjuvant chemotherapy within the first month after surgery were excluded from the analysis.

### 3.2. Comparison of Preoperative d-ROM and BAP Levels with Clinicopathological Factors in Patients with CRC

In the univariate analysis, higher d-ROM levels were significantly associated with tumor size > 45 mm, T3 or T4 tumor invasion depth, positive lymph node metastasis, and Stage II or III CRC (*p* < 0.001, <0.001, =0.004, and =0.005, respectively). The relationship between BAP values and clinicopathological factors was not significant (Table 1). The multivariate linear regression model identified tumor size as a significant explanatory variable for d-ROM level, with larger tumor sizes tending to increase d-ROM levels (Table 2). No factors were associated with BAP levels in the multivariate linear regression model (Table 3). Spearman’s correlation analysis of preoperative d-ROM and BAP values showed little correlation, with a correlation coefficient of 0.187 (*p* = 0.040) (Figure 1).

### 3.3. Comparison of Postoperative d-ROM and BAP Levels with Clinicopathological Factors in Patients with CRC

No correlation was found between postoperative d-ROM levels and clinicopathological factors in both the univariate and multivariate analyses (Appendix A). Similarly, no correlation was observed between postoperative BAP values and clinicopathological factors (Appendix A). The Spearman’s correlation coefficient analysis of postoperative d-ROM and BAP levels showed a correlation coefficient of 0.279 (*p* < 0.001), indicating a weak correlation (Appendix A).

### 3.4. Changes in d-ROM and BAP Levels Following Resection

The median preoperative d-ROM and BAP levels in 123 cases were 375 and 327, respectively. A significant decrease in d-ROM levels was observed after tumor resection (*p* < 0.001) (Figure 2a). When examining changes by stage, in stage0 and I, the median preoperative (351) and postoperative (355) d-ROM levels were not significantly different (*p* = 0.707; Figure 2b). However, in stage II, the median preoperative and postoperative d-ROM levels were 370 and 339, respectively, with a significant decrease observed (*p* = 0.03) (Figure 2c). In stage III, the median preoperative and postoperative d-ROM levels were 415 and 310, respectively, with a significant decrease observed following tumor resection (*p* < 0.001; Figure 2d). BAP levels showed no significant changes following tumor resection with respect to the cases or stages (Appendix A).

### 3.5. Comparison of Preoperative d-ROM and BAP Ratios with Clinicopathological Factors in Patients with CRC

In the univariate analysis, a higher d-ROM ratio was significantly associated with sex, tumor size > 45 mm, T3 or T4 tumor invasion depth, positive lymph node metastasis, and stage II or III (*p* = 0.018, <0.001, <0.001, <0.001, and =0.005, respectively). No relationship between the BAP ratio and clinicopathological factors was recognized (Table 4). The multivariate linear regression model identified sex, tumor size, and lymph node metastasis as significant explanatory variables for the d-ROM ratio. A large tumor size and lymph node metastasis tended to increase the d-ROM ratio (Table 5). No factors were associated with the BAP ratio in the multivariate linear regression model (Appendix A).

## 4. Discussion

Three key findings were obtained from this study. First, a reduction in the OS marker, d-ROM, was observed following tumor resection. Second, d-ROM level was correlated with tumor size. Third, although BAP level, indicating antioxidant capacity, showed a weak correlation with d-ROM levels, no significant change was observed in BAP levels following tumor resection. The increase in OS in colorectal tumor tissue was reported by Chiang et al., who measured advanced oxidation protein products and found higher levels compared to surrounding normal tissue [26], and this observation is supported by other studies [27,28,29,30,31]. The mechanisms behind the increase in OS in tumor tissue are attributed to severe inflammation observed in these tissues, where various cytokines are secreted, stimulating neutrophils and monocytes within the tumor and rapidly increasing OS [18,19]. Furthermore, mitochondrial dysfunction, metabolic changes, and frequent genetic mutations in cancer cells significantly increase ROS production, leading to the accumulation of oxidized proteins, DNA, and lipids [10,38]. Reports indicate that patients with CRC have higher OS levels compared to healthy individuals [17,38,39,40,41,42].

It has been hypothesized that the tumor microenvironment (TME), which is composed of various factors, influences systemic OS, and that removing the TME through surgery reduces systemic OS. Salehi et al. measured changes in OS markers’ MDA, ox-LDL, and AGEs before and after tumor resection in 60 patients with CRC (stages I and II) and found that these levels decreased postoperatively [38]. Similarly, Acevedo-León et al. reported a decrease in OS levels based on the disulfide form of the tripeptide L-glutamylcysteinylglycine (GSSG) in 79 patients with CRC (stages I–IV) following resection [43]. Hristozov et al. found a significant decrease in erythrocyte MDA levels postoperatively compared to preoperatively [44].

We previously reported a reduction in d-ROM levels following tumor reduction by anticancer agents [9]. These results highlight the potential of d-ROMs as a valuable biomarker for tumor burden and provide insights into the role of OS in the progression and prognosis of CRC. In our previous report, we demonstrated a correlation between blood d-ROM levels and tumor size [36]. Inokuma et al. also reported a correlation between blood d-ROM levels and tumor size [45]. In the present study, a multivariate analysis confirmed the correlation between tumor size and d-ROM levels. The preoperative d-ROM level was significantly associated with tumor size, depth of invasion (T3 or T4), lymph node metastasis, and advanced stage (II and III), consistent with previous studies reporting higher OS levels in more aggressive advanced cancers [36].

Tumor size was a significant explanatory variable for d-ROM levels in the multivariate analysis, indicating that larger tumors produce more ROS and increase systemic OS. The significant reduction in d-ROM levels following CRC resection suggests that removing the tumor, which is the main source of ROS, notably reduces systemic OS. This decrease was particularly pronounced in patients with stage II and III CRC, suggesting that tumor burden plays a crucial role in systemic OS levels. Salehi et al. reported a reduction in OS following CRC resection, but did not confirm the association between preoperative OS levels and clinicopathological factors, possibly due to the limited sample size of 60 stage I and II cases [38].

The findings on antioxidant markers following tumor resection are inconsistent, with some studies reporting a decrease and others showing an increase. For instance, Salehi et al. reported an increase in antioxidant factors after the resection of high oxidative stress (OS) tumor tissue [38], whereas Acevedo-León et al. documented a decrease in antioxidant factors following similar procedures [43]. In our study, we observed a weak correlation between d-ROM and BAP levels, suggesting that the decrease in d-ROM levels post-resection might lead to a decrease in BAP levels. However, no significant change in BAP levels was detected after surgery, despite the reduction in ROS production indicated by lower d-ROM levels. This suggests that while tumor removal reduces ROS production, the overall systemic antioxidant capacity, as measured by BAP, remains largely unaffected.

Additionally, no correlation was found between BAP levels and clinicopathological factors, in contrast to the clear correlation between d-ROM levels and tumor progression. It is plausible that BAP, as a marker of antioxidant potential, is less influenced by the tumor itself and more dependent on other physiological mechanisms, explaining the lack of change in BAP levels post-resection. Previous studies have reported that antioxidant capacity can increase when cells are stimulated by heightened OS [46], but our results suggest that strategies focusing on reducing ROS production, rather than enhancing antioxidant defenses, may be more effective in managing OS in patients with CRC.

Despite the strengths of this study, some limitations are acknowledged. First, the measurement of d-ROMs as a marker of oxidative stress may lack sufficient sensitivity in early-stage colorectal cancer, where its clinical significance appears limited. However, d-ROMs are considered to be effective in advanced colorectal cancer, and their utility in assessing treatment efficacy has also been demonstrated in stage IV cases, as previously reported in [36]. Although this study included a larger sample size compared to other reports examining changes in oxidative stress (OS) following colorectal cancer (CRC) resection, and allowed for multivariate analysis, the cohort was limited to patients from a single institution, which may restrict the generalizability of the findings. Furthermore, long-term follow-up is required to assess the sustained impact of tumor resection on OS and antioxidant capacity. Additionally, while a significant reduction in d-ROM levels was observed post-resection, the precise mechanisms driving this reduction remain unclear. Future research should focus on elucidating the molecular pathways responsible for the reduction in OS following CRC resection and explore the potential benefits of combining surgical resection with other therapeutic strategies to further mitigate OS and improve patient outcomes.

## 5. Conclusions

The physical removal of CRC, a primary source of OS, reduces blood levels of d-ROMs, a marker of OS. These results suggest that d-ROMs could serve as a valuable biomarker for monitoring tumor burden and the efficacy of surgery in patients with CRC.

## Figures and Tables

**Figure 1 cancers-16-03550-f001:**
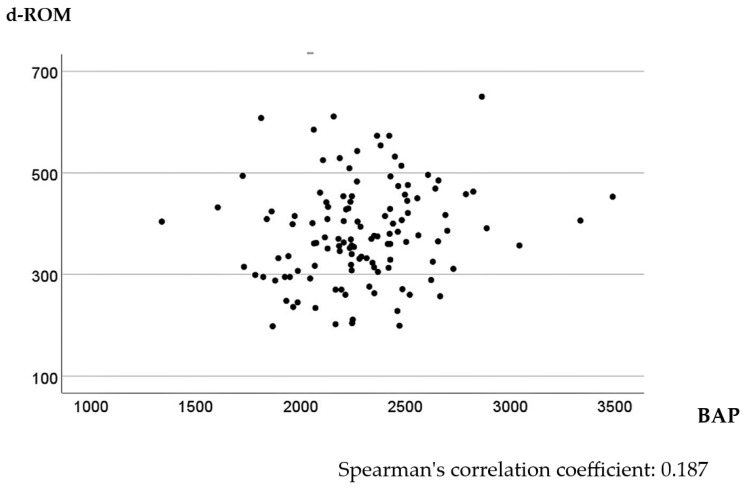
Scatter plot of d-ROMs and BAP values before surgery.

**Figure 2 cancers-16-03550-f002:**
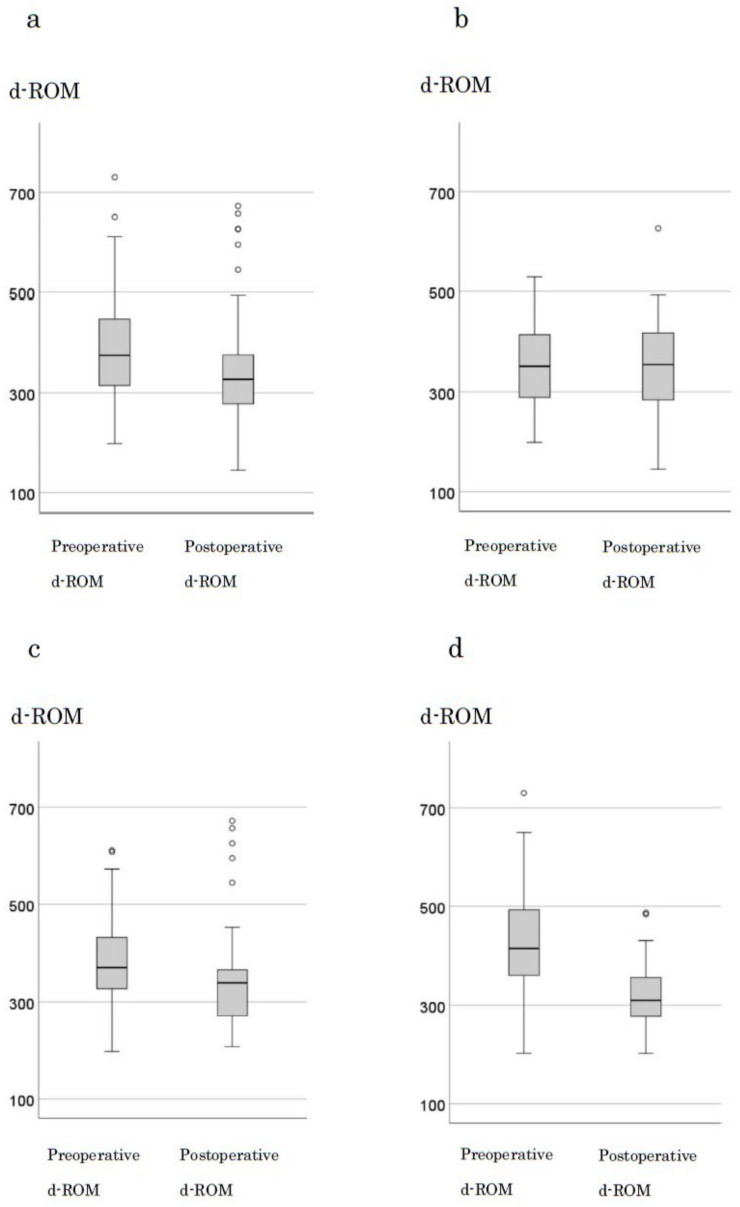
Changes in d-ROM before and after surgery by stage. (**a**) all cases (**b**) stage 0, I cases (**c**) stage III cases (**d**) stage IV cases.

**Table 1 cancers-16-03550-t001:** Comparison of preoperative d-ROM and BAP levels with clinicopathological factors in patients with colorectal cancer using the Mann–Whitney U test.

Independent Variables	N	d-ROM	*p*-Value	BAP	*p*-Value
		Median (IQR)		Median (IQR)	
Age (years)			0.428		0.897
<70	54	376.0 (311.5, 476.8)		2269.5 (2109.3, 2444.8)	
≥70	69	370.0 (319.0, 430.0)		2255.0 (2128.0, 2481.0)	
Sex			0.478		0.933
Male	66	361.0 (314.3, 439.8)		2275.0 (2128.3, 2463.5)	
Female	57	391.0 (319.0, 453.0)		2245.0 (2107.0, 2481.0)	
Location			0.063		0.613
Right	19	401.5 (349.0, 457.3)		2302.5 (2073.3, 2486.8)	
Left	104	365.0 (306.0, 429.0)		2249.0 (2125.5, 2456.0)	
Tumor size (mm)			<0.001		0.928
<45	75	357.0 (293.5, 406.0)		2246.0 (2143.5, 2477.5)	
≥45	48	426.0 (359.3, 493.3)		2275.5 (2085.8, 2454.0)	
Tumor invasion depth			<0.001		0.281
T1, T2	42	352.5 (289.8, 426.0)		2318.5 (2138.5, 2516.8)	
T3, T4	81	400.0 (332.0, 457.0)		2249.0 (2107.0, 2429.0)	
Lymph node metastasis			0.004		0.674
No	81	361.0 (306.0, 416.0)		2286.0 (2128.5, 2476.5)	
Yes	42	422.5 (359.3, 493.3)		2240.5 (2086.8, 2454.0)	
Stage			0.005		0.398
0–I	41	351.0 (289.0, 415.0)		2351.0 (2129.0, 2504.0)	
II–III	82	400.5 (332.8, 470.8)		2252.0 (2109.3, 2477.8)	

**Table 2 cancers-16-03550-t002:** Multivariate linear regression analysis results for preoperative d-ROM levels in patients with colorectal cancer.

Independent Variables	Multivariate Linear Regression Analysis		
	β	e^β^ (95%CI)	*p*-Value
Age	1.57e^−3^	0.998 (−5.58e^−3^~2.42e^−3^)	0.436
Sex (men vs. female)	3.13e^−2^	1.032 (−6.15e^−2^~1.24e^−1^)	0.505
Location (left vs. right)	4.34e^−2^	0.958 (−1.42e^−1^~5.52e^−2^)	0.385
Tumor size	4.77e^−3^	1.005 (1.87e^−3^~7.68e^−3^)	<0.001
Serosa invasion (no vs. yes)	−1.67e^−2^	0.983 (−1.35e^−1^~1.010e^−1^)	0.778
Lymph node metastasis (no vs. yes)	6.54e^−2^	1.068 (−4.15e^−2^~1.72e^−1^)	0.436

β: regression coefficient CI: confidence interval.

**Table 3 cancers-16-03550-t003:** Multivariate linear regression analysis results for preoperative BAP levels in patients with colorectal cancer.

Independent Variables	Multivariate Linear Regression Analysis		
	β	95%CI	*p*-Value
Age	−3.071	−13.404, 7.262	0.557
Sex (men vs. female)	−157.11	−396.781, 82.551	0.197
Location (left vs. right)	−232.823	−487.596, 21.949	0.072
Tumor size	−0.146	−7.644, 7.353	0.969
Serosa invasion (no vs. yes)	6.101	−298.201, 310.404	0.968
Lymph node metastasis (no vs. yes)	−48.655	−324.804, 227.495	0.728

β: regression coefficient CI: confidence interval.

**Table 4 cancers-16-03550-t004:** Comparison of clinicopathological factors with d-ROMs and BAP ratio in colorectal cancer patients using Mann–Whitney U test.

Independent Variables	Case	d-ROM Ratio	*p*-Value	BAP Ratio	*p*-Value
		Median (IQR)		Median (IQR)	
Age (years)			0.123		0.697
<70	54	0.831 (0.708, 0.986)		1.034 (0.903, 1.137)	
≥70	69	0.924 (0.765, 1.072)		1.007 (0.932, 1.148)	
Sex			0.018		0.982
Male	66	0.828 (0.682, 0.828)		1.007 (0.914, 1.138)	
Female	57	0.941 (0.781, 1.079)		1.008 (0.923, 1.148)	
Location			0.242		0.766
Right	19	0.870 (0.741, 1.020)		1.017 (0.946, 1.164)	
Left	104	0.896 (0.735, 1.103)		1.008 (0.907, 1.137)	
Tumor size (cm)			<0.001		0.754
<45	75	0.978 (0.804, 1.107)		1.008 (0.917, 1.154)	
≥45	48	0.777 (0.687, 0.891)		1.006 (0.919, 1.112)	
Tumor invasion depth			<0.001		0.599
T1, T2	42	1.037 (0.839, 1.110)		0.981 (0.905, 1.133)	
T3, T4	81	0.832 (0.711, 0.970)		1.035 (0.924, 1.140)	
Lymph node metastasis			<0.001		0.369
No	81	0.978 (0.786, 1.107)		0.990 (0.904, 1.135)	
Yes	42	0.769 (0.685, 0.897)		1.042 (0.938, 1.153)	
Stage			0.005		0.241
0–I	41	1.046 (0.883, 1.111)		0.970 (0.901, 1.111)	
II–III	82	0.829 (0.708, 0.967)		1.044 (0.926, 1.146)	

**Table 5 cancers-16-03550-t005:** Multivariate linear regression analysis results for d-ROM ratio in patients with colorectal cancer.

Independent Variables	Multivariate Linear Regression Analysis		
	β	e^β^ (95%CI)	*p*-Value
Age	2.68e^−3^	1.003 (−1.42e^−3^, 6.79e^−3^)	0.198
Sex (men vs. female)	1.27e^−1^	1.135 (3.18e^−2^, 2.22e^−1^)	0.009
Location (left vs. right)	7.58e^−2^	1.079 (−2.53e^−2^, 1.77e^−1^)	0.140
Tumor size	−3.57e^−3^	0.996 (−6.551e^−3^, −5.963e^−3^)	0.019
Serosa invasion (no vs. yes)	−2.86e^−2^	0.972 (−1.49e^−1^, 9.22e^−2^)	0.640
Lymph node metastasis (no vs. yes)	−1.13e^−1^	0.893 (−0.223, −0.003)	0.044

β: regression coefficient CI: confidence interval.

## Data Availability

All data included in this study are available from the corresponding author upon request.

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
