# Peer review of "Reduction of Blood Oxidative Stress Following Colorectal Cancer Resection"

_cancers, 2024, doi:10.3390/cancers16203550_

Round 1

Reviewer 1 Report

Comments and Suggestions for Authors

In the manuscript, authors were trying to draw a connection between reactive oxygen metabolite derivatives (d-ROM) and tumor burden or surgical efficacy, by comparing the d-ROM levels and biological antioxidant potentials (BAP) before and after the surgery, the authors found that the d-ROM levels are associated with the tumor size and d-ROM levels reduce following tumor resection, suggesting the d-ROM levels could serve as marker for tumor burden. However, there are several flaws in this study.

1. the authors detected the BAP along with the d-ROM levels, but these two factors are not consistent, the authors need to explain this inconsistence. 

2. the correlation between BAP and d-ROM in Figure 1 seems off, should be redo the analysis.

3. the authors include the analysis of tumor location in their study, I am wondering what is the rational for this, is there any evidence showed the location of colorectal cancer impacts the progression or survival?

4. the authors claim that d-ROM could serve as marker for surgical efficacy, but it is only showed the remarkable difference in high stage tumors, and the maximum reduction is about 22%, which is big range to evaluate during clinic.

5. In figure2, authors used different Y axis values which may mislead the readers, the figures should be redraw and use the same standards.

Comments on the Quality of English Language

minor issues

Author Response

1.the authors detected the BAP along with the d-ROM levels, but these two factors are not consistent, the authors need to explain this inconsistence.

(Response)

Thank you for your insightful comment regarding the inconsistency between BAP and d-ROM levels. We agree that this is an important point to address, and we have revised the relevant section of the Discussion (lines 251–268) to clarify this issue.

In our study, we observed that while reactive oxygen species (ROS) production, as indicated by d-ROM levels, decreased following colorectal cancer resection, there was no significant change in systemic antioxidant capacity, as measured by BAP levels. This can be explained by the fact that, although d-ROM levels were found to correlate with tumor progression, no such correlation was observed between BAP levels and clinicopathological factors. We hypothesize that BAP, as a marker of antioxidant potential, is less influenced by the tumor itself and more reliant on other physiological mechanisms, thus potentially explaining the lack of significant change in BAP levels despite reduced oxidative stress.

In response to your comment, we have modified the text from line 254 to line 271 of the Discussion and highlighted the revised portion in yellow for easy reference. We hope these changes provide a clearer explanation for the observed inconsistency and address your concerns.

Thank you once again for your valuable feedback.

2.the correlation between BAP and d-ROM in Figure 1 seems off, should be redo the analysis.

(Response)

Thank you for your valuable feedback regarding the correlation between BAP and d-ROM in Figure 1. In response to your comment, we have reanalyzed the data and made necessary adjustments to both the figure and the corresponding text in the manuscript.

3. the authors include the analysis of tumor location in their study, I am wondering what is the rational for this, is there any evidence showed the location of colorectal cancer impacts the progression or survival?

(Response)

Thank you for your insightful question regarding the inclusion of tumor location analysis in our study. We are grateful for the opportunity to clarify this point.

Recent evidence suggests that the location of colorectal cancer can indeed impact both progression and survival outcomes. For instance, a systematic review and meta-analysis published in JAMA Oncology (Feb 2017, Vol 3, Issue 2) titled "Prognostic Survival Associated With Left-Sided vs Right-Sided Colon Cancer" reported that patients with left-sided colorectal cancer had a significantly reduced risk of death compared to those with right-sided tumors. This finding highlights the potential prognostic significance of tumor location.

Furthermore, the colorectal cancer treatment guidelines also emphasize that the efficacy of chemotherapy and molecular targeted therapies may vary based on tumor location. Specifically, different treatment strategies are recommended for right-sided versus left-sided colorectal cancers, particularly regarding the use of molecular targeted therapies. This is why we considered tumor location as an important factor in our analysis, as it may influence both treatment outcomes and overall prognosis.

We hope this addresses your concern and are happy to provide further clarification if needed.

4. the authors claim that d-ROM could serve as marker for surgical efficacy, but it is only showed the remarkable difference in high stage tumors, and the maximum reduction is about 22%, which is big range to evaluate during clinic.

(Response)

We greatly appreciate your insightful feedback on our manuscript. As you rightly noted, the utility of d-ROMs as a marker for surgical efficacy appears to be more relevant in advanced-stage colorectal cancer. In early-stage cancer, where the risk of recurrence is lower, the role of d-ROMs as a marker may indeed be limited. We believe that the significance of d-ROM lies primarily in its application to advanced cancers, where monitoring tumor burden and recurrence risk is more critical.

Moreover, unlike traditional tumor markers such as CEA, oxidative stress markers like d-ROM have a tendency to reach a saturation point, beyond which their values do not increase proportionally. This results in a constrained dynamic range for evaluating changes, which may limit their applicability in clinical settings. Nevertheless, as referenced in our previous study (Reference 36), we investigated the changes in oxidative stress in relation to chemotherapy efficacy in stage IV colorectal cancer. In that study, d-ROM demonstrated high sensitivity and specificity, indicating its potential as a tumor marker that complements CEA.

In response to your comment, we have revised the Limitations section of the manuscript, particularly in the 273th to 278th lines, to explicitly state that the value of d-ROM as a tumor marker may be less significant in early-stage colorectal cancer. The added text has been highlighted in yellow for ease of reference.5.

5.In figure2, authors used different Y axis values which may mislead the readers, the figures should be redraw and use the same standards.

 (Response)

We acknowledge the inconsistency in the Y-axis values of Figure 2, which could potentially cause confusion for the readers. To address this, we have redrawn the figure using uniform Y-axis values across all panels to ensure clarity and consistency. We believe this revision improves the readability of the figure and aligns with your suggestion.

Please find the revised Figure 2 included in the updated manuscript.

Thank you again for your constructive feedback, which has helped improve the quality of our paper.

Reviewer 2 Report

Comments and Suggestions for Authors

The manuscript title “Reduction of Blood Oxidative Stress Following Colorectal Cancer Resection” by Sawai et al; is an interesting article. However, after reading it thoroughly, I have some concerns that need to be addressed properly for the betterment of the manuscript quality. See below;

Comment1: Given the high heterogeneity of human colorectal cancer, relying solely on reactive oxygen metabolite derivatives (d-ROMs) and biologic antioxidant potential (BAP) findings may limit the study's generalizability. To enhance the manuscript's quality, the authors may consider providing inflammatory and non-inflammatory cytokine profiling data, which would facilitate a better understanding of the correlation between colorectal cancer resection and d-ROMs levels.

Comment2: Considering the association between tumor size and decreased d-ROMs levels following tumor resection (mentioned in the discussion section), could the authors elaborate on the implications for colorectal cancer recurrence patients? 

Furthermore, would it be feasible to provide d-ROMs and cytokine profiling data for patients who underwent resection and subsequently experienced recurrence, to further explore this relationship?

Comment3: In the discussion section, could the authors elaborate on the association between d-ROMs and colorectal cancer stem cells (CSCs), exploring the relationship between oxidative stress and CSC markers' expression (e.g., CD44, CD133, ALDH1)? This would provide valuable insight for the broader journal audience, highlighting the significance of CSCs in colorectal cancer.

Comment4: Could you increase the font size of the numbers in Figure 1& 2 for better readability?

Author Response

1.Given the high heterogeneity of human colorectal cancer, relying solely on reactive oxygen metabolite derivatives (d-ROMs) and biologic antioxidant potential (BAP) findings may limit the study's generalizability. To enhance the manuscript's quality, the authors may consider providing inflammatory and non-inflammatory cytokine profiling data, which would facilitate a better understanding of the correlation between colorectal cancer resection and d-ROMs levels.

(Response)

We thank the reviewer for the careful review of the manuscript. In our previous research, we have explored the relationship between d-ROM levels and the inflammatory marker neutrophil-to-lymphocyte ratio (NLR), as described in reference 36. Our findings demonstrated that there is no direct correlation between d-ROMs and NLR. However, both d-ROMs and NLR have proven to be valuable independent markers for assessing tumor burden, and when used in combination, they may improve prognostic predictions.

While we did not include cytokine profiling in the current study, we recognize its importance and plan to investigate the impact of changes in both d-ROMs and inflammatory responses on prognosis in future research.

2.Considering the association between tumor size and decreased d-ROMs levels following tumor resection (mentioned in the discussion section), could the authors elaborate on the implications for colorectal cancer recurrence patients? 

Furthermore, would it be feasible to provide d-ROMs and cytokine profiling data for patients who underwent resection and subsequently experienced recurrence, to further explore this relationship?

(Response)

We thank the reviewer for the careful review of the manuscript. Among the 123 patients, 20 experienced recurrence. We created ROC curves based on preoperative and postoperative d-ROMs and BAP values to evaluate recurrence risk, but Kaplan-Meier analysis did not show significant differences between d-ROMs, BAP values, and recurrence-free survival (RFS).

The graphs are shown below. We believe that extending the observation period and increasing the number of cases will provide more accurate insights. Additionally, analyzing d-ROMs and cytokine data in patients with recurrence is a future task.

3.In the discussion section, could the authors elaborate on the association between d-ROMs and colorectal cancer stem cells (CSCs), exploring the relationship between oxidative stress and CSC markers' expression (e.g., CD44, CD133, ALDH1)? This would provide valuable insight for the broader journal audience, highlighting the significance of CSCs in colorectal cancer.

(Response)

We greatly appreciate the reviewer’s insightful comment. CD44v9, a variant isoform of CD44, is indeed known to interact with glutamate-cysteine transporters to enhance intracellular glutathione (GSH) levels, thereby mitigating oxidative stress and supporting the survival of cancer stem cells (CSCs). Our previous work, published in Cancers (2024 Apr 19;16(8):1556. ), highlighted the significance of CD44v9 expression in both primary tumor tissues and circulating tumor cells as a poor prognostic factor in colorectal cancer.

At present, however, there are no published reports directly addressing the impact of blood oxidative stress levels on the expression of CSC markers such as CD44 or on the efficacy of chemotherapy in this context. Additionally, the relationship between CSC reduction following tumor resection and changes in blood oxidative stress markers has not been thoroughly studied. Therefore, while we recognize the potential importance of this area, the absence of direct evidence limits our ability to speculate further on these mechanisms within the scope of this study.

We believe this is an important topic for future research and plan to investigate the link between oxidative stress markers, CSC dynamics, and therapeutic outcomes in colorectal cancer in subsequent studies.

We hope this response is satisfactory and look forward to your feedback.

4.Could you increase the font size of the numbers in Figure 1& 2 for better readability?

(Response)

We appreciate the reviewer’s helpful suggestion. In response, we have increased the font size of the numbers in Figures 1 and 2 to enhance readability. The updated figures are included in the revised manuscript for your review.

We hope this improves the clarity of the figures and look forward to any further feedback.

Round 2

Reviewer 1 Report

Comments and Suggestions for Authors

The authors have addressed all my concerns, but the figures need to be edited, which would allow it to be more Intuitive. 

Author Response

The authors have addressed all my concerns, but the figures need to be edited, which would allow it to be more Intuitive. 

(response)

Thank you for your valuable feedback. I have revised the figures as per your suggestion. Specifically, I have adjusted the Y-axis of Figure 1 to match the axis values of Figure 2, ensuring consistency and making the data presentation clearer and more intuitive. I hope this addresses your concern, and I appreciate your thoughtful comments.

Please let me know if any further revisions are needed.